# EdgeNet: An End-to-End Deep Neural Network Pretrained with Synthetic Data for a Real-World Autonomous Driving Application

**DOI:** 10.3390/s25010089

**Published:** 2024-12-27

**Authors:** Leanne Miller, Pedro J. Navarro, Francisca Rosique

**Affiliations:** División de Sistemas e Ingeniería Electrónica (DSIE), Campus Muralla del Mar, s/n, Universidad Politécnica de Cartagena, 30202 Cartagena, Spain; leanne.miller@upct.es (L.M.); paqui.rosique@upct.es (F.R.)

**Keywords:** end-to-end architectures, multimodal synthetic dataset, autonomous driving

## Abstract

This paper presents a novel end-to-end architecture based on edge detection for autonomous driving. The architecture has been designed to bridge the domain gap between synthetic and real-world images for end-to-end autonomous driving applications and includes custom edge detection layers before the Efficient Net convolutional module. To train the architecture, RGB and depth images were used together with inertial data as inputs to predict the driving speed and steering wheel angle. To pretrain the architecture, a synthetic multimodal dataset for autonomous driving applications was created. The dataset includes driving data from 100 diverse weather and traffic scenarios, gathered from multiple sensors including cameras and an IMU as well as from vehicle control variables. The results show that including edge detection layers in the architecture improves performance for transfer learning when using synthetic and real-world data. In addition, pretraining with synthetic data reduces training time and enhances model performance when using real-world data.

## 1. Introduction

End-to-end neural networks are evolving rapidly in artificial intelligence and have become particularly popular in the field of autonomous driving. Traditional autonomous driving systems are modular in nature, wherein the driving problem is divided into subtasks operating independently such as object detection [1,2], localization [3], trajectory planning [4], and control [5]. While this approach allows for specialized task optimization, it also introduces significant drawbacks, particularly in terms of error propagation between modules and computational inefficiencies [6]. The end-to-end approach, also known as direct perception, offers substantial benefits, such as reduced complexity [7], decreased error propagation [8], and improved computational efficiency [9], making it an appealing solution for autonomous driving applications.

End-to-end architectures learn driving-related behaviors, resulting in more integrated and efficient autonomous driving systems [10]. These architectures enable the mapping of raw sensor inputs from the perception system directly to control commands for autonomously driving a vehicle.

This work presents a state-of-the-art end-to-end architecture for autonomous driving based on the EfficientNetV2 architecture obtaining high performance and efficiency. The architecture has been designed to bridge the domain gap between synthetic and real-world images for end-to-end autonomous driving applications and includes novel edge detection layers before the convolutional module for this purpose. The architecture has been pretrained using a multimodal synthetic dataset: Carla Multimodal Raw Data (CarlaMRD), created using the Carla simulator. The pretrained architecture has then been fine-tuned for real-world driving applications using the UPCT real-world dataset, obtaining promising results whilst reducing the computation time.

## 2. Related Work

Different approaches to solving the end-to-end problem exist: while most studies have focused on generating outputs directly, others have incorporated intermediate steps such as trajectory planning. For example, in [11], the authors addressed autonomous navigation by predicting future waypoints, using a front-facing RGB camera and maps, as an intermediate step between the input and control output. Other studies have combined trajectory planning with direct perception, as seen in research using the Carla simulator. In the work by Wu et al. [7], the authors used an architecture with two branches: one branch predicted the trajectory while the other focused on the control variables. The trajectory branch used encoder data to guide the control predictions, and the outputs from both branches were then fused.

In addition to different architectural approaches, the use of multimodal inputs is another key factor for improving the accuracy and robustness of end-to-end models [12,13]. By combining data from multiple sensors, such as cameras, IMU, LiDAR, and RADAR, these models can achieve a more comprehensive understanding of the environment. For instance, the TransFuser [14] architecture is based on transformers and fuses image and LiDAR data using self-attention layers. In this architecture, multiple transformer modules operate at different resolutions to merge features from the input data, which are then fed into a waypoint prediction architecture.

Expanding on the advantages of multimodal inputs, the choice of specific data types, such as RGB images and depth maps, has also been shown to significantly enhance the performance of end-to-end architectures, particularly those utilizing convolutional layers for feature extraction [15]. RGB images provide detailed visual information about the driving environment while depth maps offer crucial spatial and distance data, helping interpret the three-dimensional structure of the scene [16]. By fusing RGB images with depth maps, architectures can better perceive and navigate complex environments, as demonstrated by the work of Xiao et al. [17].

Synthetic data has gained popularity as a valuable resource for end-to-end applications, primarily due to the challenges of collecting high-quality real-world data. One of the main advantages of synthetic data is the ability to generate large datasets for training purposes. Although synthetic data can constitute a powerful tool, end-to-end architectures trained exclusively in simulated environments struggle to perform well in real-world scenarios without adaptation [18]. Transfer learning allows models initially trained in simulations to be fine-tuned using real-world data, helping bridge the gap between the two domains and enhancing the versatility of architectures across different environments and conditions. However, most current research on transfer learning focuses on specific tasks such as object detection [19,20] or reducing the domain gap [21,22], rather than extending to more complex applications like end-to-end driving systems, where models must directly output control commands for vehicles.

## 3. Materials and Methods

### 3.1. Synthetic Dataset

The rapid advance in autonomous driving technology has increased the demand for high-quality datasets to train, validate, and test end-to-end architectures for predicting vehicle control variables. These datasets are crucial for the development of robust autonomous driving systems capable of navigating diverse and complex real-world scenarios. Traditionally, real-world datasets collected using sensor-equipped vehicles have been the foundations for developing these systems. However, collecting real-world data is very expensive and time-consuming, and it is often difficult to obtain large amounts of data for training purposes [23], causing interest in synthetic datasets as a viable alternative to increase in recent years [24].

Synthetic datasets contain artificially generated data that mimic the conditions and scenarios of real-world driving environments. These datasets are created using computer-generated imagery and physics-based simulations, offering a controlled environment where diverse driving situations can be systematically produced and manipulated [25]. This capability addresses several limitations of real-world datasets such as variations in weather and lighting conditions [26], complicated traffic situations [27], and the need for extensive data labeling [28]. In addition, adverse and potentially dangerous weather conditions such as fog, waterlogged roads, snow, and hail can be simulated, enabling a model to train in these situations, aiming to avoid the model detecting these conditions erroneously [29].

The evaluation of current synthetic datasets for perception and vehicle control reveals notable limitations and areas for improvement. Most existing synthetic datasets have been designed for very specific applications and include only the necessary sensors and data for performing certain tasks such as odometry [30] and object detection [31] or for specific weather conditions [32]. In Table 1, a summary of the most popular synthetic driving datasets is presented, including sample sizes, perception sensors, and vehicle control data and whether the datasets include raw data or processed data. The most significant multimodal datasets are included, and although not recent, the Udacity dataset is included as it was considered one of the most complete multimodal datasets [33].

Despite the large number of existing studies, most autonomous driving datasets provide labelled data primarily for classification tasks, such as semantic segmentation, rather than raw sensor data [27,28,34,35,36,37,38]. This limitation restricts the development and evaluation of end-to-end perception models [39], which rely on raw sensor inputs to directly predict driving actions or scene understanding without intermediate steps like segmentation [40]. Moreover, the public datasets identified lacked crucial driving control variables such as steering wheel angle, acceleration, and vehicle speed. These variables are essential for the effective training and performance of end-to-end architectures. It was also observed that many synthetic datasets have small sample sizes, considering that one of the advantages of simulators is the possibility of generating large amounts of synthetic data [35,36,37,38].

In this work, CarlaMRD dataset has been designed, collecting raw data from various sensors in different weather and traffic scenarios, and can be adapted for use in different types of applications. The synthetic dataset has been developed using the Carla simulator. Carla is an open-source simulator for autonomous driving research designed to support development, training, and validation of autonomous urban driving systems. The simulation platform provides different urban scenarios, with buildings, vehicles, pedestrians, etc., and supports flexible specification of sensor suites and environmental conditions [41].

#### 3.1.1. Simulation Setup

Before running simulations in Carla and recording data, it is important to correctly configure the simulation. First, a map must be chosen or created from scratch. For this dataset, six predefined Carla worlds have been chosen (Figure 1):Town01 is a small simple residential village.Town02 is a simple town with a mixture of residential and commercial buildings.Town03 is a medium-sized urban map with junctions and a roundabout.Town04 is a small mountain village with a highway in the shape of an infinity sign.Town05 is a squared-grid town with cross junctions and a bridge. It has multiple lanes in each direction to perform lane changes.Town10 is a larger urban environment with skyscrapers and residential buildings.

The weather and time of day can also be modified. This can be achieved by custom-defining each weather variable or choosing from one of fourteen predefined settings. In this work, some custom weather conditions have been defined in addition to using the predefined functions. The adjustable weather and time of day variables are the following: (a) cloudiness, (b) precipitation, (c) sun altitude angle, (d) sun azimuth angle, (e) precipitation deposits, (f) wind intensity, (g) fog density, and (h) wetness.

In addition to the geographical and meteorological variables of the simulation, the traffic conditions can also be defined. For each of the simulations, vehicles have been added to create realistic traffic scenarios. A wide range of vehicles can be added, from different car models to trucks and emergency vehicles. For the simulations, a black Ford Lincoln MKZ model vehicle has been chosen. Pedestrians and cyclists also play an important role in autonomous driving tasks and have been added to the simulations for increased realism.

To gather the data from the driving simulations, a range of perception sensors have been configured onboard the simulated vehicle. The simulated sensors in Carla can be configured with similar variables to real-world sensors such as resolution and field of view (fov) for cameras or range and number of channels for LiDAR sensors. To position the sensors onboard the vehicle, a Carla transform consisting of 3D coordinates is used, with origin in the center of the vehicle. The RGB and depth cameras have been situated above the front windscreen for a frontal view while the LiDAR and IMU sensors have been positioned on the middle of the roof of the vehicle. The RADAR sensor has been positioned on the front of the vehicle above the bumper. The sensors used and their configuration for recording data during the simulations are shown in Table 2.

#### 3.1.2. Simulation Design

To create the dataset, 100 simulations have been run, each with a duration of six minutes of driving time, with the vehicle set to autopilot mode. The advantage of using lots of short simulations is that a larger variety of scenes can be recorded, combining different weather conditions and traffic situations. The vehicle is randomly spawned to one of the spawn points on the maps and is driven around the simulation scene, recording data from the sensors and vehicle control. To ensure accurate data synchronization and a precise temporal sequence, timestamp values have been recorded with the data for each sensor. Figure 2 shows some examples of the images captured by the RGB camera.

The data collected from each of the 100 simulations are saved to a directory. Inside this folder, the RGB and depth images are each saved in their respective directories. The data collected from the sensors (IMU, RADAR, LiDAR, and vehicle control) are saved to an individual text file in csv format for each sensor.

Once all of the simulations have been run, the data are reviewed to make sure that all the files have been saved correctly and that the simulation has finished without any incidents. The timestamps from each of the different sensors have been synchronized by the simulator; therefore, for each sensor, a data sample exists for that timestamp, without the need for manual synchronization, which is the case when using real-world data [42].

### 3.2. End-to-End Architecture Design

End-to-end architectures have played a significant role in the advance of deep learning, especially in the field of computer vision. In recent years, architectures containing convolutional layers have become the preferred architectures for tasks such as image classification, object detection, and segmentation due to their ability to extract meaningful patterns from visual data.

End-to-end architectures consist of sequences of specialized layers that work together to transform the input data to obtain the desired results: in this case, the control actions of the vehicle. These layers include convolutional layers that apply filters to detect local patterns in the image data, dense fully connected layers that use the features extracted to make the final predictions, and pooling layers that are commonly used to reduce the spatial dimensions of the data while preserving key features [9]. Additionally, custom layers can be used for applying specific functions to the input data. In end-to-end architectures, these layers are usually stacked multiple times to deepen the ability of the network to extract increasingly complex features. To introduce non-linearity to the architecture and reduce vanishing gradient issues, non-linear activations like ReLU (Rectified Linear Unit) are commonly used in CNNs.

The end-to-end architecture proposed in this work is based on the EfficientNetV2B0 model, designed and implemented to efficiently handle both visual and inertial input data. This dual input structure is based on the architecture designed in the work by Navarro et al. [9], which found that incorporating additional input data improved results substantially. The model incorporates edge detection layers along with an EfficientNetV2B0 backbone, which is responsible for feature extraction from RGB and depth images. Additionally, the architecture includes dense layers specifically designed to process data from an Inertial Measurement Unit (IMU), integrating multiple sensor modalities.

In the work by Navarro et al. [9], both acceleration and angular velocity were tested as additional inputs, and it was concluded that angular velocity enhanced performance in the prediction of both the speed and steering angle outputs. Therefore, in this work, angular velocity has been used as the additional input.

The EfficientNetV2 architecture belongs to a family of models that range from B0, optimized for smaller images, to B3, suited for larger images. This family of models has consistently outperformed traditional CNNs whilst at the same time offering an improved computational speed and efficiency in terms of parameters [43]. The EfficientNetV2 models are particularly noted for their ability to achieve better performance on smaller datasets compared to Vision Transformer models, which typically require larger datasets to achieve the best results [44]. For example, the EfficientNetV2B0 model comprises 7.4 million parameters, significantly fewer than some Vision Transformer models, which can contain up to 304 million parameters. Furthermore, Vision Transformer models demand over 24 times more training time compared to the EfficientNetV2B0 architecture [43], making the latter a more practical choice for many applications, particularly when computational resources are limited. Another advantage of the EfficientNetV2 architectures is their flexibility in training. They can either be trained from scratch or leverage pretrained weights from large datasets like ImageNet.

The EfficientNetV2B0 model was adapted and trained from scratch in this work; this was necessary due to the inclusion of depth images as an additional channel alongside the standard RGB input, resulting in four-channel images. The pretrained ImageNet weights, designed for three-channel RGB images, were therefore incompatible with this extended input structure.

In the proposed architecture, the output of the EfficientNetV2B0 block has been merged with a second input branch, enabling the integration of additional sensor information such as linear acceleration, angular velocity, vehicle orientation, or even GPS data. While RGB and depth images serve as the primary inputs, IMU sensor data, specifically angular velocity, are utilized as secondary inputs. This multimodal input capability enhances the model’s ability to process and interpret complex environments. Additional output layers have also been incorporated into the architecture, increasing the total number of parameters to 7.67 million. An illustration of the architecture is provided in Figure 3, demonstrating how the different data streams are integrated into the CNN.

Before the images are fed into the EfficientNetV2B0 architecture, they undergo a series of pre-processing steps designed to enhance critical features using edge detection. This process is implemented through four Lambda layers, which emphasize areas in the image where significant intensity changes occur, typically marking object boundaries. These edge features help the model focus on important visual cues that are essential for driving-related tasks such as identifying lanes, detecting obstacles, and recognizing nearby vehicles. By extracting these edges, the model becomes better equipped to concentrate on the most relevant aspects of the scene, improving its accuracy in interpreting complex driving environments.

The first two Lambda layers in the pre-processing pipeline handle basic input extraction, separating the RGB channels and the depth channel from the input image. The third Lambda layer applies edge detection specifically to the RGB channels using the Sobel operator, a widely used method for highlighting areas of rapid intensity change. The result is a set of edge maps that highlight key boundaries in the image. The fourth Lambda layer then normalizes the edge maps, ensuring consistent scaling and contrast for the model’s input. To further prepare the data for the EfficientNetV2B0 block, a Rescaling layer is used to standardize the pixel values.

Once pre-processing is complete, the images are passed to the EfficientNetV2B0 block, which contains 240 layers. This architecture is composed of various types of layers, including convolutional layers, batch normalization layers, and specialized operations organized into blocks. The efficient design of these blocks enables the model to process the image data effectively while maintaining a balance between speed and accuracy.

In addition to the image input, the architecture also processes IMU data, which are handled separately using two dense layers. These layers transform the IMU data, in this case angular velocity, into a format that can be integrated with the visual data. The outputs of the EfficientNetV2B0 block and the processed IMU data are then merged using a Concatenate layer, combining the visual and sensor inputs.

At the output stage, the architecture splits into two branches, each responsible for predicting one of the output variables. These branches predict vehicle speed (Km/h) and steering angle (radians). Each one of the output branches consists of three dense layers, followed by a final dense layer with a size of one, corresponding to the single value predicted for each output variable. This branching design allows the model to make independent predictions for both vehicle speed and steering angle, leveraging the combined information from the RGB images, depth images, and IMU data to deliver accurate outputs. In the study performed by Navarro et al. [9], it was found that using a branch for each output variable obtained better results than using one vector output.

## 4. Results

### 4.1. Model Configuration

The end-to-end architecture has been used to validate the synthetic CarlaMRD dataset. The architecture has been designed and implemented using the Tensorflow 2.10 and Keras 2.10 libraries. The models were trained on a PC with an NVIDIA GeForce RTX 3070 GPU. To train the model, 150,850 samples from the synthetic dataset were used (120 × 160 RGB and depth images, angular velocity in °/s from the IMU, and the vehicle control parameters, speed in km/h, and steering angle in radians were used). The hyperparameters applied are shown in Table 3. A batch size of 20 was chosen to balance GPU memory usage and training stability. This value was determined empirically after testing with smaller and larger batch sizes. The RMSprop optimization function [45] was proven to achieve the best performance and accuracy.

To avoid overfitting during the training of the models, a stop condition that considered the validation metrics was used. A patience of 10 epochs was used after which, if the model was no longer learning, training was stopped and the weights from the best epoch were restored. This method ensured that the model had finished training without overfitting occurring.

To split the data, the K-Fold cross method was used. The dataset was split into six equal sets of 25,141 samples, with five for training and testing each of the five folds and one for validation. As a result, five models were obtained, validated using the same set of validation data to obtain consistent results. The five models were then tested using the corresponding test set for each fold; this way, predictions were obtained for a larger amount of data, providing a better idea of the performance of the model.

Three metrics were calculated to evaluate the performance of the models. These metrics were chosen to gain a thorough view of how the model behaved and allow the results to be compared with those presented by other authors in the literature:The Mean Absolute Error (MAE), which is the average of the absolute differences between the predicted and actual values.The Mean Absolute Percentage Error (MAPE), which is calculated by dividing the MAE by the range of the speed and angle data. The range is the difference between the maximum and minimum values of the variables to predict.The coefficient of determination, R^2^, which was used to evaluate the quality of the results obtained by the model.

To obtain a global view of the models and their accuracy, the metrics were calculated for all the test predictions from each of the five models, obtaining a total of 125,705 predictions. The models took, on average, 28,778 s to train, with 64 epochs. The overall results and those results obtained from each of the five models are shown in Table 4.

The results show that the models achieved a lower percentage error for detecting the steering angle compared to the vehicle speed. This is logical as it is usually easier to relate geometrical features such as the road lines rather than spatial information to calculate the speed, especially as the models apply edge detection to the RGB images before the convolutional layers of the Efficient Net block. However, regarding the coefficient of determination, the models appeared to make better predictions for the speed variable.

To study the quality of the predictions, box and whisker plots for the speed and angle errors are shown in Figure 4. The median error of the speed prediction was close to zero with a value of 0.081 Km/h and half of the errors had a value between −0.567 and 0.903 Km/h. For the angle predictions, the median error was negative at −0.058°, with the first quartile at −0.238° and the third quartile at 0.062°. For both the speed and angle predictions, the error predictions took on a Gaussian distribution.

### 4.2. Application of the Pretrained CNN for Training with a Real-World Dataset

#### 4.2.1. Real-World Dataset

To study the usability of the synthetic dataset in real-world applications, the pretrained model was tested with a real driving dataset. For this application, the UPCT dataset containing real-world driving data was chosen [23]. The UPCT dataset is a public dataset that contains multimodal data from a variety of perception sensors including 3D LiDAR, RGB and depth cameras, IMU, GPS, encoders, and biometric data from drivers. The data were recorded with state-of-the-art equipment onboard the UPCT’s CICar autonomous vehicle. The UPCT dataset contains 78,000 samples that were obtained by a group of 30 different drivers performing tests along an urban route in southern Spain with real traffic, including roundabouts, junctions, merging traffic situations, and street parking. An example of some scenes included in the UPCT dataset is shown in Figure 5. The tests were performed at different times of day, including the morning, afternoon, and early evening.

#### 4.2.2. Baseline Training with Only Real-World Data

First, the CNN model from Section 3.2 was trained using only the UPCT dataset to verify the performance of the model using real-world data. The data were split into six equal groups of 13,000 data samples wherein five were used for training and testing with the K-fold method. The last group was used for the validation for each of the five models in the K-fold method, to obtain consistent results, and to have more data samples for testing.

The metrics were calculated for all the predictions from each of the five models, obtaining a total of 65,000 predictions. The models took, on average, 19,880 s to train, with 62 epochs. The results from each fold and the overall predictions are shown in Table 5.

The results obtained by the models for predicting the vehicle speed using the real-world dataset were promising and had improved compared to those achieved with the synthetic dataset. The angle predictions, however, did not gain a significant improvement. As with the synthetic dataset, the angle predictions achieved a lower percentage error compared to the speed predictions.

Box and whisker plots for the speed and angle errors are shown in Figure 6. For the speed prediction errors, the median was close to zero with a value of −0.009 Km/h and the first and third quartile values were −0.315 Km/h and 0.290 Km/h, respectively. As shown by the box and whisker plots, the errors for both the speed and angle predictions took on a Gaussian distribution with the median in the center of the box. The median for the angle prediction errors was positive in this case with a value of 0.107°. The first and third quartile values were between −0.124° and −0.355°, respectively.

#### 4.2.3. Pretraining with the Synthetic Dataset

The third test consisted of performing transfer learning using the synthetic dataset to pretrain the models with the aim of reducing the training time when training with real-world data. After training the models with the synthetic data, the weights from the Efficient Net convolutional blocks were saved and loaded to the models before training with the real dataset.

The metrics were calculated for all predictions from the test sets each of the five models, obtaining a total of 65,000 predictions. The models took, on average, 17,350 s to train, with 56 epochs. The results from each fold and the overall results obtained from transfer learning are shown in Table 6.

From the results, it can be observed that pretraining a model with a synthetic dataset and using the weights to train real data decreases the training time needed to obtain the same results. In this work, the models needed, on average, six epochs less for training with the same real-world dataset compared to training with no previous information. Box and whisker plots of the prediction error values are shown in Figure 7.

The prediction errors calculated with the pretrained models were very similar to those with no pretraining, once again with a Gaussian distribution with narrow boxes centered around the median value. For the speed prediction errors, a median of −0.040 Km/h was obtained, as were quartile values between −0.461 Km/h and 0.376 Km/h. The median value of the angle prediction errors in this case was negative with a value of −0.037° and the first and third quartiles had values of −0.306° and 0.217°, respectively.

### 4.3. Analysis of the Architecture with and Without Edge Detection Layers for Transfer Learning

Finally, the EdgeNet architecture was tested, removing the edge detection layers from the architecture, to study the impact that these layers had on the performance of the architecture for transfer learning applications. The tests were repeated using the same datasets divided into five folds with the same training, validation, and test sets and the same hyperparameters and stopping condition being used. Table 7 shows the results obtained by the architecture both with and without the edge detection block for both the synthetic and real-world datasets, as well as for pretraining the architecture with the synthetic dataset to be used with real-world data.

It can be observed that when using the synthetic and real-world datasets alone, the architecture achieved a high performance with and without the edge detection block. However, the edge detection block was proven to enhance the performance of the architecture significantly when pretraining with synthetic data. The edge detection block reduced the domain gap between the synthetic data and real-world data scenarios, and with the pretrained weights, a similar performance was achieved using real-world data with a lower computational cost than training from scratch. The effectiveness of the pretrained weights was largely attributed to the diversity of the synthetic dataset, which captured a wide range of lighting, traffic, and weather scenarios. This variability enhanced the dataset’s ability to generalize across different real-world conditions.

In the case of the speed prediction, with the edge detection layers, the MAPE was reduced to almost a third compared to the architecture without edge detection layers. Box and whisker plots of the MAE values for the transfer learning predictions without the edge detection block are shown in Figure 8.

The box and whisker plots for the architecture without edge detection layers (Figure 8) had a normal distribution similar to that of the architecture with edge detection layers (Figure 7). It could be observed that the interquartile range (IQR) of the MAE values for the architecture without the edge detection block was greater for the prediction of both the speed and the angle variables. In the case of the speed MAEs, without the edge detection block, the errors had a significantly larger dispersion. Table 8 includes the numerical values of the median, quartile, and IQR values of the MAEs of the architectures with and without the edge detection block.

## 5. Discussion

The results obtained by the EdgeNet architecture were compared to those presented by other authors in the literature. Several studies existed, mainly using real-world ad hoc datasets obtained or modified by the authors. Some authors used synthetic datasets and real-world datasets for training their architectures [46,47]. However, these were conducted as separate experiments, and the synthetic data were not used for pretraining the models. Table 9 shows a comparison of the results obtained in this work with other experiments using CNN models for end-to-end driving and details on the types of datasets used and the variables predicted.

One of the first end-to-end deep neural network applications for autonomous driving was resented by Bojarski et al. [48]. They used the PilotNet model for object detection and steering angle prediction from just RGB image inputs from the Udacity dataset. The work by Yang et al. [46] built on these results and used the PilotNet to predict both the vehicle speed and steering angle from the same Udacity RGB images. Xu et al. tested six different CNN models, some using LSTM architectures, to predict the steering angle using 21,000 short 36 s videos of RGB frames as training data. They used the real-world BDDV dataset, and the best accuracy was obtained with a temporal CNN with an accuracy of 84.6% [49]. In the work presented by Wang et al. [47], different end-to-end deep convolutional neural networks were tested to predict the speed and steering angle using RGB images as the input. It is worth noting that a better performance was obtained using the model with the synthetic dataset compared to using the real-world data. The authors in [9] completed a thorough study of three types of CNNs with different inputs, one with RGB images, a second complementing the RGB images with IMU data, and a third model using sequences of RGB images. The best results were achieved using the RGB images with an additional input of IMU data, obtaining a MAPE of 1.69% for the speed calculation and a MAPE of 0.43% for predicting the steering wheel angle. The experiment conducted by Prasad et al. [50] presented an end-to-end CNN model to predict the steering angle from real-world RGB images on a small vehicle; the only metric given was the R^2^ score with a value of 0.819.

The end-to-end model presented in this work obtained a 99.30% accuracy for speed calculation and a 99.49% accuracy for predicting the steering angle, with the real-world dataset used without pretrained weights. When using the model with the pretrained weights from training with the synthetic dataset, the model obtained a 98.95% accuracy predicting speed and a 99.47% accuracy predicting the steering angle with the real-world dataset. In addition, the results show that by pretraining a CNN model with synthetic data, the training time can be significantly reduced.

## 6. Conclusions

End-to-end architectures trained and tested using only simulated driving data have shown promising results. However, few approaches have focused on addressing the gap between simulation and reality and the benefits that transfer learning applications have to offer. In this work, an end-to-end architecture was developed, which not only obtained a high performance with simulated data and real-world data alone but also showed significant potential when used for transfer learning. To overcome the differences between simulated and real-world images, edge detection layers were introduced into the architecture before the EfficientNetV2 module. These layers extract crucial edge information before the convolutional module, helping bridge the domain gap between synthetic and real-world data. The architecture was evaluated using two datasets: a simulated dataset generated by the Carla simulator and the real-world UPCT dataset.

The proposed architecture integrates both RGB and depth images as inputs, along with a second input branch for inertial data, to enhance accuracy and performance. The architecture was trained and tested in three scenarios: (1) with the CarlaMRD synthetic dataset, (2) with the real-world UPCT dataset, and (3) with the UPCT dataset using the pretrained weights from the synthetic dataset. The results obtained with the architecture including the edge detection block were then compared to those obtained by the same architecture without the edge detection. It was proven that the inclusion of edge detection layers significantly improved the predictions for the speed variable when using an architecture pretrained with synthetic data.

The results show that pretraining with the synthetic dataset significantly reduces the training time using weights pretrained with synthetic data when training with real-world data. Furthermore, the architecture obtained a high performance and computational efficiency in predicting vehicle control variables whether pretraining was used or not. The results achieved demonstrate a notable improvement compared to similar studies conducted by other authors, highlighting the effectiveness and robustness of the proposed architecture. However, one drawback of this architecture is that it is limited to predicting the vehicle control variables, and while this information is very important, its application may be limited when the vehicle must interact with other drivers in dense traffic situations. In addition, the architecture relies on the quality of the synthetic dataset and its ability to represent real-world scenarios. In less structured environments such as unmarked rural roads, further exploration may be required.

Using data from multiple sensors, such as the IMU, improves performance as it provides complementary information. While the RGBD images capture visual and spatial information, the IMU data aid in understanding the vehicle’s movement and orientation. This work is a continuation of the paper by Navarro et al. [9], where different data modalities were tested. The authors concluded that adding inertial data, specifically angular velocity, to the architecture significantly enhanced performance.

Future work could address several areas to enhance the EdgeNet architecture. One development could be the incorporation of different datasets during pretraining to improve domain generalization. Additionally, using data augmentation techniques that mimic real-world changes, like lens flare or sensor noise, can make the system more reliable and robust.

## Figures and Tables

**Figure 1 sensors-25-00089-f001:**
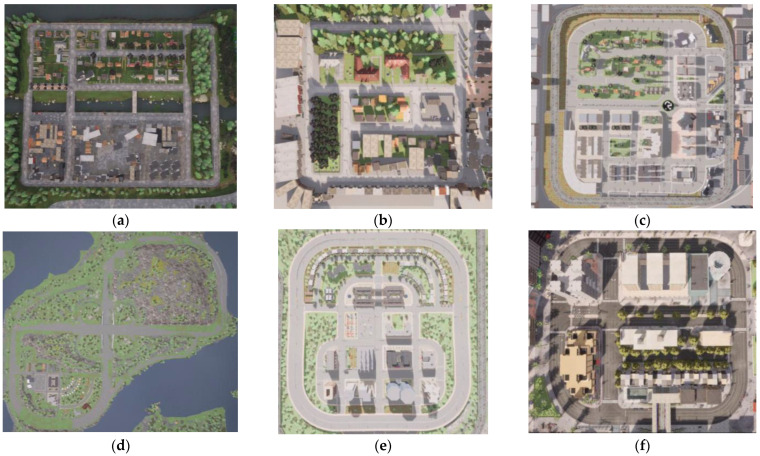
Carla world maps: (**a**) Town01; (**b**) Town02; (**c**) Town03; (**d**) Town04; (**e**) Town05; (**f**) Town10.

**Figure 2 sensors-25-00089-f002:**
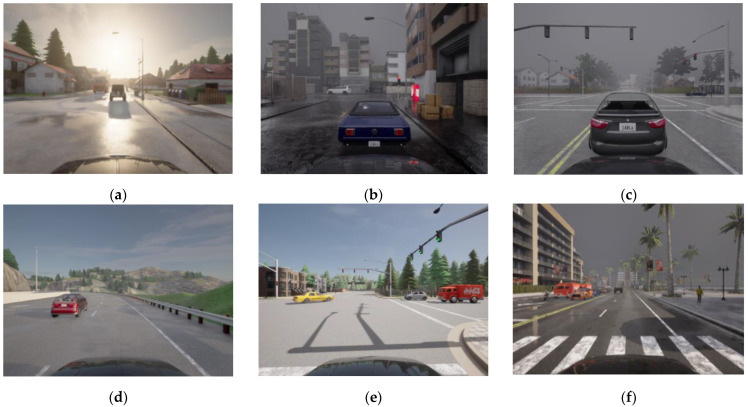
Example RGB images: (**a**) Town01: wet sunset; (**b**) Town02: heavy rain noon; (**c**) Town03: foggy afternoon; (**d**) Town04: sunny afternoon; (**e**) Town05: sunny noon; (**f**) Town10: cloudy evening.

**Figure 3 sensors-25-00089-f003:**
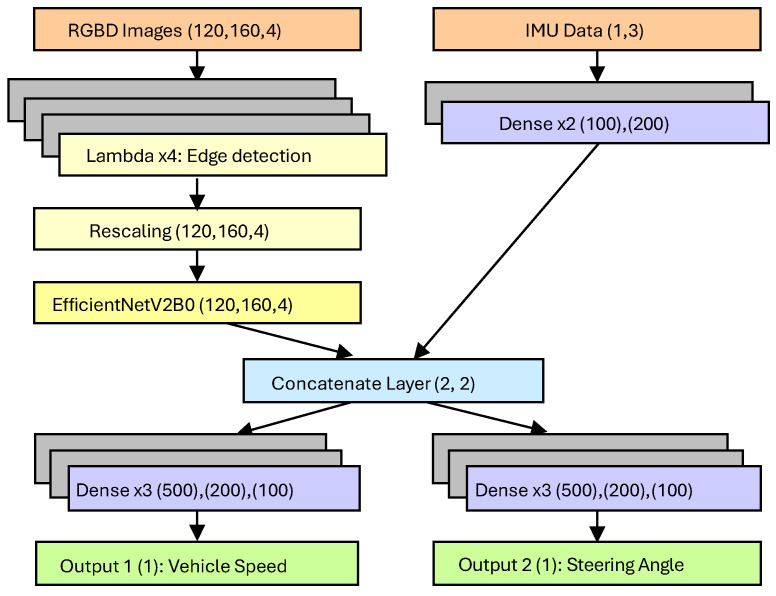
End-to-end architecture with EfficientNetV2 B0 backbone.

**Figure 4 sensors-25-00089-f004:**
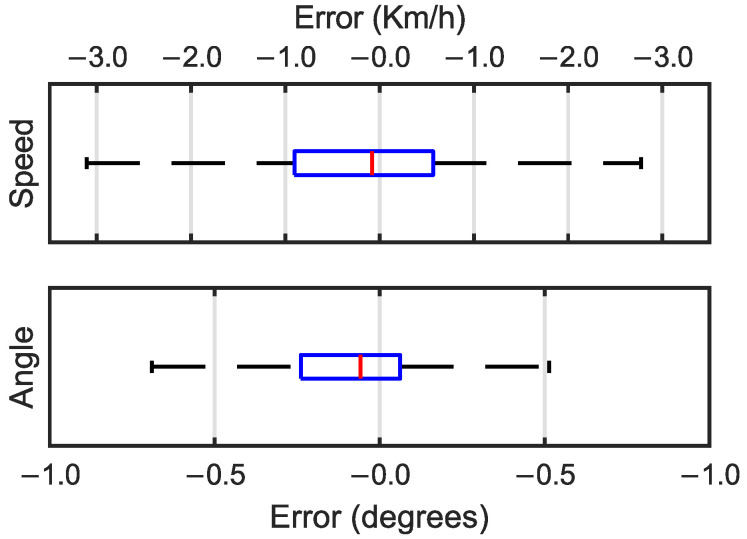
Box and whisker plots for speed errors in Km/h (**top**) and angle errors in degrees (**bottom**).

**Figure 5 sensors-25-00089-f005:**
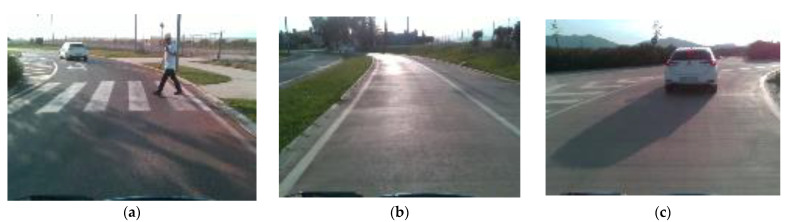
Example images from the UPCT dataset: (**a**) pedestrian crossing; (**b**) saturation due to reflections on the road; (**c**) car braking [23].

**Figure 6 sensors-25-00089-f006:**
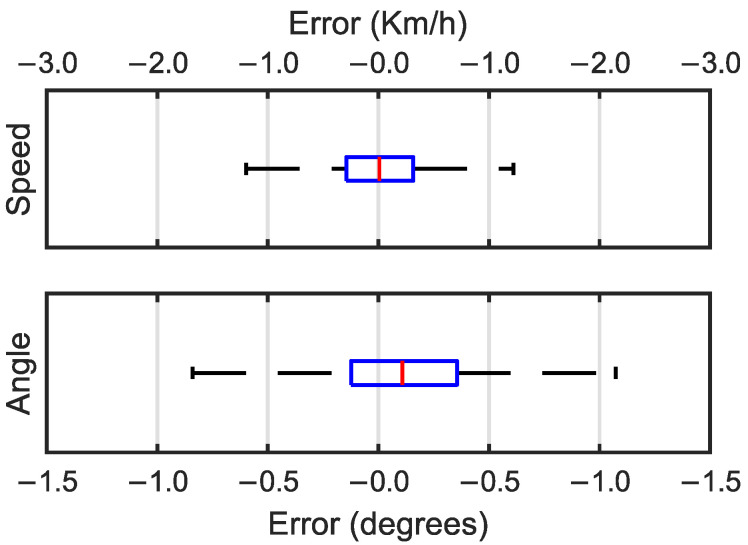
Box and whisker plots for speed errors in Km/h (**top**) and angle errors in degrees (**bottom**).

**Figure 7 sensors-25-00089-f007:**
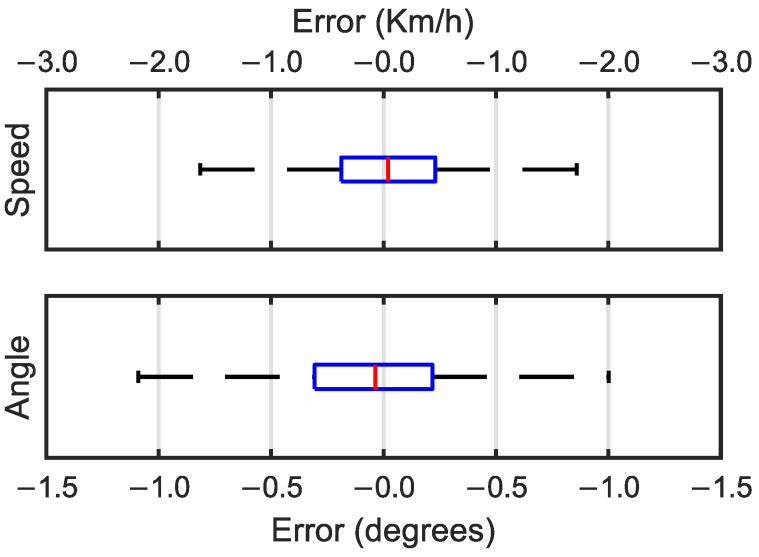
Box and whisker plots for speed errors in Km/h (**top**) and angle errors in degrees (**bottom**).

**Figure 8 sensors-25-00089-f008:**
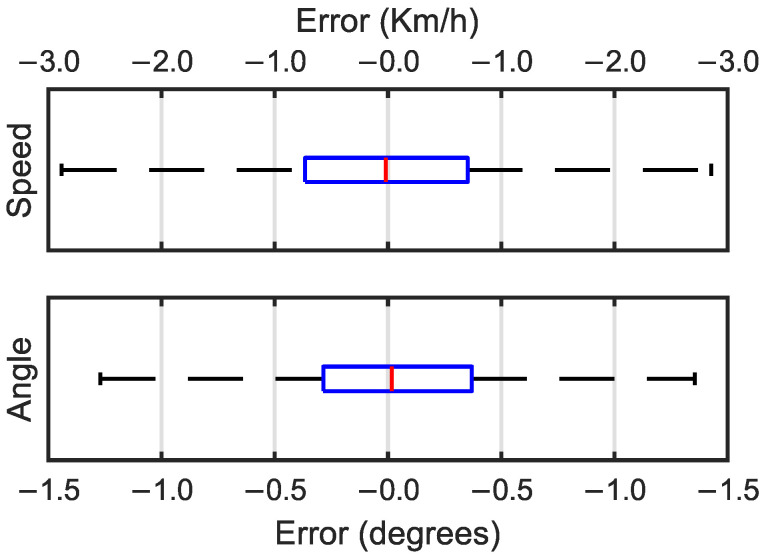
Box and whisker plots for speed errors in Km/h (**top**) and angle errors in degrees (**bottom**).

**Table 1 sensors-25-00089-t001:** Synthetic datasets with sensors for perception, vehicle control variables, and data types.

Dataset/Year	Samples	Image Type	IMU	LiDAR	RADAR	Vehicle Control	Raw Data
Udacity [33]/2016	34 K	RGB	Yes	Yes	No	Yes	Yes
SYNTHIA [34]/2016	213K	RGB	No	No	No	No	No
VEIS [35]/2018	61K	RGB	No	No	No	No	No
ParallelEye [36]/2019	40 K	RGB	No	No	No	No	No
PreSIL 6 [37]/2019	50 K	RGB	No	Yes	No	No	No
IDDA [28]/2020	1 M	RGB, D	No	No	No	No	No
CarlaScenes [30]/2021	-	RGB, D	Yes	Yes	No	No	Partial
SHIFT [27]/2022	2.6 M	RGB, D	Yes	No	No	No	No
AmodalSynthDrive [38]/2024	60 K	RGB 360°	Yes	Yes	No	No	No
Proposed: CarlaMRD	150 K	RGB, D	Yes	Yes	Yes	Yes	Yes

**Table 2 sensors-25-00089-t002:** Sensors used in the perception system and vehicle control variables.

Sensor	Data Type
RGB Camera	RGB image 640 × 480, fov = 90°.
Depth Camera	Depth image 640 × 480, fov = 90°.
IMU	Acceleration x, y, z (m/s^2^). Angular velocity x, y, z (°/s). Orientation x, y, z (°).
LiDAR	3D pointcloud, x, y, z, intensity. Channels = 64, f = 20, range = 100 m, points per second = 500,000.
RADAR	2D pointcloud: polar coordinates, distance, and velocity. Horizontal fov = 45°, vertical fov = 30°.
Vehicle Control	Steering angle (rad). Speed (km/h). Accelerator pedal (value from 0 to 1). Brake pedals (value from 0 to 1).

**Table 3 sensors-25-00089-t003:** Configuration of hyperparameters.

Parameter	Variable
Batch size	20
Optimization algorithm	RMSprop
Loss function	Huber
Metric	Mean Absolute Error
Learning rate	0.001

**Table 4 sensors-25-00089-t004:** MAE, MAPE, and R^2^ values obtained using the synthetic dataset.

Fold	Variable	MAE (Km/h, °)	MAPE	R^2^
1	Speed	1.66	1.80%	0.973
	Angle	0.65	0.71%	0.944
2	Speed	1.21	1.32%	0.986
	Angle	0.41	0.46%	0.952
3	Speed	1.41	1.53%	0.978
	Angle	0.45	0.50%	0.952
4	Speed	1.80	1.95%	0.971
	Angle	0.96	0.62%	0.939
5	Speed	1.27	1.37%	0.981
	Angle	0.45	0.49%	0.954
Overall	Speed	1.47	1.59%	0.978
	Angle	0.51	0.55%	0.948

**Table 5 sensors-25-00089-t005:** MAE, MAPE, and R^2^ values obtained using real-world data.

Fold	Variable	MAE (Km/h, °)	MAPE	R^2^
1	Speed	0.39	0.67%	0.996
	Angle	0.34	0.44%	0.985
2	Speed	0.48	0.82%	0.995
	Angle	0.49	0.62%	0.953
3	Speed	0.34	0.58%	0.997
	Angle	0.34	0.44%	0.984
4	Speed	0.44	0.75%	0.995
	Angle	0.39	0.50%	0.969
5	Speed	0.38	0.66%	0.996
	Angle	0.40	0.51%	0.977
Overall	Speed	0.40	0.69%	0.996
	Angle	0.39	0.50%	0.974

**Table 6 sensors-25-00089-t006:** MAE, MAPE, and R^2^ values obtained using pretrained weights to train with real-world data.

Fold	Variable	MAE (Km/h, °)	MAPE	R^2^
1	Speed	0.68	1.17%	0.987
	Angle	0.43	0.55%	0.969
2	Speed	0.54	0.94%	0.992
	Angle	0.36	0.47%	0.976
3	Speed	0.52	0.90%	0.992
	Angle	0.39	0.50%	0.978
4	Speed	0.70	1.21%	0.987
	Angle	0.50	0.64%	0.961
5	Speed	0.58	0.99%	0.989
	Angle	0.38	0.49%	0.981
Overall	Speed	0.61	1.04%	0.989
	Angle	0.41	0.53%	0.973

**Table 7 sensors-25-00089-t007:** Overall MAE, MAPE, and R^2^ results obtained by the architecture with and without the edge detection block.

Dataset	Variable	MAE (Km/h, °)		MAPE		R^2^	
		w Edges	w/o Edges	w Edges	w/o Edges	w Edges	w/o Edges
CarlaMRD	Speed	1.47	0.40	1.59%	0.69%	0.978	0.996
	Angle	0.50	0.39	0.55%	0.50%	0.948	0.974
UPCT	Speed	0.40	0.39	0.69%	0.68%	0.996	0.995
	Angle	0.39	0.39	0.50%	0.51%	0.974	0.972
UPCT pretrained	Speed	0.60	1.66	1.04%	2.86%	0.989	0.911
	Angle	0.41	0.60	0.52%	0.77%	0.973	0.929

**Table 8 sensors-25-00089-t008:** Median and quartile MAE values with and without the edge detection block.

Dataset	Variable	Q1		Q2 (Median)		Q3		IQR	
		w Edges	w/o Edges	w Edges	w/o Edges	w Edges	w/o Edges	w Edges	w/o Edges
UPCT pretrained	Speed (Km/h)	−0.461	−0.703	−0.040	0.019	0.375	0.731	0.836	1.434
	Angle (°)	−0.306	−0.285	−0.037	0.017	0.217	0.371	0.523	0.656

**Table 9 sensors-25-00089-t009:** Comparison of the proposed model with the metrics of other end-to-end models.

Authors, Ref.	Dataset	Data Type	Input	Output	MAE (km/h)/(°)	MAPE (%)	R^2^
Bojarski et al. [48]	Udacity	Synthetic	RGB	Steering angle	4.26	-	-
Yang et al. [46]	Udacity	Synthetic	RGB	Speed/steering angle	0.68/1.26	-	-
	SAIC	Real	RGB	Speed	1.62	-	-
Xu et al. [49]	BDDV	Real	RGB	Steering angle	-	15.4	-
Wang et al. [47]	GAC	Real	RGB	Speed/steering angle	4.25/3.55	-	-
	GTAV	Synthetic	RGB	Speed/steering angle	3.28/2.84	-	-
Navarro et al. [9]	UPCT	Real	RGB + IMU	Speed/steering angle	0.98/3.61	1.69/0.43	-
Prasad [50]	-	Real	RGB	Steering angle	-	-	0.819
Proposed: BorderNet	Carla	Synthetic	RGBD + IMU	Speed/steering angle	1.47/0.51	1.59/0.55	0.977/0.948
	UPCT	Real-world	RGBD + IMU	Speed/steering angle	0.61/0.41	1.04/0.53	0.989/0.973

## Data Availability

The dataset presented is available on the Zenodo platform: https://zenodo.org/uploads/11193178 (accessed on 21 December 2024).

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
