# Peer review of "EdgeNet: An End-to-End Deep Neural Network Pretrained with Synthetic Data for a Real-World Autonomous Driving Application"

_sensors, 2024, doi:10.3390/s25010089_

Round 1

Reviewer 1 Report

Comments and Suggestions for Authors

This paper uses Carla to generate simulation data for autonomous driving, conducts pre-training on the EfficientNet network, applies the pre-trained network weights to real data, and adds an edge detection module to improve the overall prediction effect. This paper has good innovation and engineering value. However, it is hoped that the author can pay attention to the following related issues:

(1) Lines 278 - 282: For the EfficientNet network framework, if IMU input is added, what are the specific differences between the input layer and RGBD? Please emphasize this in the text. Additionally, does the IMU only input three-dimensional angular velocity information? Is there no acceleration information? That is to say, is the output of vehicle speed only directly generated based on the image level? Please add an analysis and explanation.

(2) Lines 407 - 412: Why do the weight parameters of the EfficientNet using the generated data have good adaptability and compatibility with real data? That is to say, the simulation data is equally effective for training the network, meaning that what is trained with the generated data can be carried over to real data? Please explain the reason.

(3) Lines 440 - 442: Does the author mean that after adding edge detection, on the generated dataset, using the pre-trained weights can achieve better performance, even almost as good as using the real dataset? And it can also reduce the training burden. Please analyze the reason for this. Why does the addition of the detection module actually reduce the training time and get closer to the real data?

(4) Lines 485 - 488: The IMU data and the image data should be of different modalities. After adding the IMU data, the prediction performance of the network as a whole can be improved. Could the author analyze the reason for this conclusion? Is it that multiple sensors can inherently improve the performance of the end-to-end network? If possible, please give an explaination of that.

Author Response

Reviewer #1 Response to Questions:

We would like to thank the reviewer for the comments and suggestions, as these have contributed to enhancing the overall quality of our paper.

  • Lines 278 - 282: For the EfficientNet network framework, if IMU input is added, what are the specific differences between the input layer and RGBD? Please emphasize this in the text. Additionally, does the IMU only input three-dimensional angular velocity information? Is there no acceleration information? That is to say, is the output of vehicle speed only directly generated based on the image level? Please add an analysis and explanation.

The EfficientNet network is designed for image processing tasks (such as identification and classification). For this reason, it is only used in the RGBD branch and not with the IMU input.

The IMU sensor supplies XYX accelerations and XYZ angular velocities among others. This work follows the work by Navarro et al. (reference 9 in the manuscript), where it was concluded that the angular velocity information enhanced both the vehicle speed and the vehicle steering angle more than the linear acceleration data when using real world data. The tests were repeated with synthetic data and similar conclusions were reached, for this reason the angular velocity was used. This has been added and explained in the manuscript in lines 229 - 232.

  • Lines 407 - 412: Why do the weight parameters of the EfficientNet using the generated data have good adaptability and compatibility with real data? That is to say, the simulation data is equally effective for training the network, meaning that what is trained with the generated data can be carried over to real data? Please explain the reason.

The EdgeNet architecture uses edge detection layers to detect the edges in the images before these are fed to the efficientnet block. The aim was to find extractable features common to both the simulated and real-world images, to try to bridge the domain gap. A more detailed explanation is given in section 4.3 where the results are compared to those without the edge detection layers (lines 443 – 452).

The effectiveness of the pre-trained weights is largely attributed to the diversity of the synthetic dataset, which captures a wide range of lighting, traffic, and weather scenarios. This variability enhances the dataset's ability to generalize across different real-world conditions.

  • Lines 440 - 442: Does the author mean that after adding edge detection, on the generated dataset, using the pre-trained weights can achieve better performance, even almost as good as using the real dataset? And it can also reduce the training burden. Please analyze the reason for this. Why does the addition of the detection module actually reduce the training time and get closer to the real data?

With this architecture, we aimed to reduce the domain gap between synthetic images and real-world images. By extracting the edges, the differences between real and generated images such as lighting and colours do not affect the model as greatly. The model generally trains in fewer epochs using the pretrained weights as it starts training closer to the solution than without the pretrained weights.

  • Lines 485 - 488: The IMU data and the image data should be of different modalities. After adding the IMU data, the prediction performance of the network as a whole can be improved. Could the author analyze the reason for this conclusion? Is it that multiple sensors can inherently improve the performance of the end-to-end network? If possible, please give an explaination of that.

Using data from multiple sensors, such as the IMU, improves performance because it provides complementary information. While the RGBD images capture visual and spatial information, the IMU data aids in understanding the vehicle's movement and orientation. This work is a continuation of the paper in reference 9, where different data modalities were tested and it was concluded that adding inertial data, specifically angular velocity, to the architecture significantly enhanced performance.

This explanation has been added to the manuscript in the conclusion section, lines 543 - 548.

Reviewer 2 Report

Comments and Suggestions for Authors

This paper suggests an end-to-end architecture for autonomous driving that leverages the EfficientNetV2 backbone. The proposed architecture is designed to bridge the domain gap between synthetic and real-world images. To enhance feature extraction, edge detection layers are integrated before the convolutional module.

The paper is nice and I enjoyed reading it; however, I have several concerns:

1. The Introduction and the related work section should be split into two sections. One long section that incorporates so much information is undesirable.

2. The description of Figure 1(d) is indistinct. I would rephrase it to "Highway in the shape of an infinity sign".

3. In Figure 3, the caption "Dense x3 (500),(200),(100)" appears twice. Is there a difference between the rectangles? If yes, please write it in the figure. If no, why have two rectangles?

4. In the configuration of hyperparameters (table 3), why did the authors use a batch size of 20? A rationalization is needed.

5. The comparison made in Table 4 and Table 5 is very important; however, why did the authors put it in 2 tables? The data would be easier to understand if it were presented in a single graph.

6. The previous comment also applies to the pair of tables 6,7 and the pair of tables 8,9.

7. In Sekkat, A. R., Mohan, R., Sawade, O., Matthes, E., & Valada, A., "Amodalsynthdrive: A synthetic amodal perception dataset for autonomous driving", IEEE Robotics and Automation Letters, Vol. 9(11), pp. 9597-9604, 2024, the authors suggest a synthetic model perception dataset for autonomous driving. I would encourage the authors to discuss pros and cons of their system compared to this system.

8. In Wiseman Y., "Real-Time Monitoring of Traffic Congestions", IEEE International Conference on Electro Information Technology, Lincoln, Nebraska, USA, pp. 501-505, 2017, available online at: https://u.cs.biu.ac.il/~wisemay/eit2017.pdf  , the author gives examples in Figure 5 and Figure 6 of his paper for false alarm in vehicles, where the vehicles can detect splashed water or hail as rigid objects. I would encourage the author to cite this paper and discuss how they can handle splashed water or hail in the road.

9. A discussion of the potential limitations and future improvements of the proposed model would be beneficial.

10. The format of references should be consistent.

Author Response

Reviewer #2 Response to Questions:

We would like to thank the reviewer for the comments and suggestions, as these have contributed to enhancing the overall quality of our paper.

  1. The Introduction and the related work section should be split into two sections. One long section that incorporates so much information is undesirable.

Ok, we have separated the introduction and related work into two sections.

  1. The description of Figure 1(d) is indistinct. I would rephrase it to "Highway in the shape of an infinity sign".

Ok, we have changed the description of Figure 1(d) in lines 146-147.

  1. In Figure 3, the caption "Dense x3 (500),(200),(100)" appears twice. Is there a difference between the rectangles? If yes, please write it in the figure. If no, why have two rectangles?

Figure 3 represents the architecture layout. The output consists of two independent branches in parallel, one for each output. The block is repeated as this group of layers are present in both output branches.

  1. In the configuration of hyperparameters (table 3), why did the authors use a batch size of 20? A rationalization is needed.

A batch size of 20 was chosen to balance GPU memory usage and training stability. This value was determined empirically after testing with smaller and larger batch sizes. This has been included in lines 309 – 311.

  1. The comparison made in Table 4 and Table 5 is very important; however, why did the authors put it in 2 tables? The data would be easier to understand if it were presented in a single graph.

We agree with this suggestion, the tables have been combined into a single table to present the results more clearly.

  1. The previous comment also applies to the pair of tables 6,7 and the pair of tables 8,9. These results tables have also been combined.

  1. In Sekkat, A. R., Mohan, R., Sawade, O., Matthes, E., & Valada, A., "Amodalsynthdrive: A synthetic amodal perception dataset for autonomous driving", IEEE Robotics and Automation Letters, Vol. 9(11), pp. 9597-9604, 2024, the authors suggest a synthetic model perception dataset for autonomous driving. I would encourage the authors to discuss pros and cons of their system compared to this system.

We have cited the suggested paper adding it to Table 1, where existing datasets and their characteristics are presented for comparison. This dataset, as well as others included in Table 1, lacks vehicle control data, which is necessary for the end-to-end learning architecture presented in this work (lines 118 – 128).

  1. In Wiseman Y., "Real-Time Monitoring of Traffic Congestions", IEEE International Conference on Electro Information Technology, Lincoln, Nebraska, USA, pp. 501-505, 2017, available online at: https://u.cs.biu.ac.il/~wisemay/eit2017.pdf , the author gives examples in Figure 5 and Figure 6 of his paper for false alarm in vehicles, where the vehicles can detect splashed water or hail as rigid objects. I would encourage the author to cite this paper and discuss how they can handle splashed water or hail in the road.

We have cited the suggested paper, explaining how these conditions can be simulated to help the models train and react correctly if it encounters harsh weather conditions in lines 103-106.

  1. A discussion of the potential limitations and future improvements of the proposed model would be beneficial.

We agree. In the conclusion section we have included a new paragraph detailing future improvements and some limitations (lines 549 - 553). Potential drawbacks of the architecture have also been included in lines 537 - 542.

  1. The format of references should be consistent.

We have revised the format of the references and corrected this where needed.

Round 2

Reviewer 2 Report

Comments and Suggestions for Authors

The authors made a decent effort and the paper is certainly publishable so I would recommend accepting the paper.

Author Response

Dear Reviewer,

Thank you for your constructive feedback on our paper. We appreciate your positive assessment and are pleased to hear that you find the paper publishable.